# Endosomal Sorting Protein SNX27 and Its Emerging Roles in Human Cancers

**DOI:** 10.3390/cancers15010070

**Published:** 2022-12-22

**Authors:** Shreya Deb, Jun Sun

**Affiliations:** 1Division of Gastroenterology and Hepatology, Department of Medicine, University of Illinois at Chicago, Chicago, IL 60612, USA; 2Department of Microbiology and Immunology, University of Illinois at Chicago, Chicago, IL 60612, USA; 3University of Illinois at Chicago (UIC) Cancer Center, University of Illinois at Chicago, Chicago, IL 60612, USA; 4Jesse Brown VA Medical Center, Chicago, IL 60612, USA

**Keywords:** autophagy, colon cancer, endosomal pathway, endocytic recycling, sorting nexin, transmembrane proteins, tight junctions, tumorigenesis

## Abstract

**Simple Summary:**

The sorting nexin (SNX) family of proteins mediate the sorting and trafficking of endocytosed transmembrane proteins between the endosomal compartments, lysosome, trans-Golgi network, and plasma membrane. SNXs are characterized by the presence of a Phox domain; however, SNX27 is a unique member of this family as it contains an additional PDZ domain. Evidence suggests that SNX27, in association with the retromer complex, binds cargo via its PDZ domain and recycles them from the early endosomes to the plasma membrane, thereby escaping their lysosomal degradation. The roles of SNX27 in cancer is largely unexplored, however many of its identified cargoes have been indicated in tumorigenesis. In this review we provide a summary of the current knowledge, as supported by scientific literature and evidence, regarding the fundamental structure, biological function, and implication of SNX27 in human cancers.

**Abstract:**

SNX27 belongs to the sorting nexin (SNX) family of proteins that play a critical role in protein sorting and trafficking in the endocytosis pathway. This protein family is characterized by the presence of a Phox (PX) domain; however, SNX27 is unique in containing an additional PDZ domain. Recently, SNX27 has gained popularity as an important sorting protein that is associated with the retromer complex and mediates the recycling of internalized proteins from endosomes to the plasma membrane in a PDZ domain-dependent manner. Over 100 cell surface proteins have been identified as binding partners of the SNX27–retromer complex. However, the roles and underlying mechanisms governed by SNX27 in tumorigenesis remains to be poorly understood. Many of its known binding partners include several G-protein coupled receptors, such as β2-andrenergic receptor and parathyroid hormone receptor, are associated with multiple pathways implicated in oncogenic signaling and tumorigenesis. Additionally, SNX27 mediates the recycling of GLUT1 and the activation of mTORC1, both of which can regulate intracellular energy balance and promote cell survival and proliferation under conditions of nutrient deprivation. In this review, we summarize the structure and fundamental roles of SNX proteins, with a focus on SNX27, and provide the current evidence indicating towards the role of SNX27 in human cancers. We also discuss the gap in the field and future direction of SNX27 research. Insights into the emerging roles and mechanism of SNX27 in cancers will provide better development strategies to prevent and treat tumorigenesis.

## 1. Introduction

SNX27 belongs to the large family of SNX proteins whose fundamental role involves the regulation of intracellular trafficking, as well as sorting of internalized cargo through the complicated cascade of the endocytic pathway [1]. The SNX proteins are mainly characterized by the presence of a specific phospholipid binding Phox (PX) homology domain [2]. However, the presence or absence of additional domains further categorize the SNX proteins into different subgroups as they regulate distinct functions in target binding and specificity [3]. This includes the SNX-BAR subfamily that consists of an additional BAR (Bin/Amphiphysin/Rvs) domain [4] and the SNX-FERM subfamily with additional FERM (4.1/ezrin/radixin/moesin) domain [1]. SNX27 belongs to the SNX-FERM subfamily. However, structural analysis has revealed that in addition to the PX and FERM domains, SNX27 carries a third PDZ (post-synaptic density 95/discs large/zonula occludens-1) binding domain. This makes SNX27 the most unique member of the SNX family because it mediates the rescue of endocytosed transmembrane proteins from lysosomal degradation in a PDZ-domain dependent manner [5].

Endocytosis is a crucial cellular process that internalizes cargoes from the cell surface and transports them in membrane-bound vesicles for sorting into either of its two destined fates: lysosomal degradation or recycling back to the plasma membrane [6]. Improper sorting of internalized cargo and defects in the endosomal recycling pathway have been linked to a variety of human diseases, such as neurodegeneration disorders as well as cancer [7]. However, the underlying mechanisms governing endosomal cargo sorting, retrieval and recycling, and its implication in disease pathophysiology is still poorly understood. 

Retromer is an endosomal coat protein complex first identified in the endolysosomal system of budding yeast [8]. It is evolutionarily conserved in mammals and has been implicated in the sorting and transport of several transmembrane proteins from early endosomes to either the trans-Golgi network (TGN) or the cell surface, thus rescuing the cargo from lysosomal degradation [9]. The mammalian retromer complex consists of a constitutive heterotrimer of VPS26-VPS29-VPS35 proteins that has been shown to assist in cargo recognition [10]. Several members of the SNX family of proteins have been identified to associate with the retromer complex, such as the SNX1/2 and SNX5/6 heterodimer of the SNX-BAR subfamily [4]; the monomeric SNX3 of the SNX-PX subfamily [11]; and the most recently identified associate, SNX27 [1]. While the SNX-BAR heterodimers assist in retromer-mediated endosomal membrane remodeling [10], SNX3 and SNX27 have been suggested to act as cargo adaptors [1]. However, the association of SNX27 and the retromer complex is critical for the cell surface recycling of transmembrane proteins carrying a PDZ-binding motif [3]. Therefore, the emerging roles of the SNX27-retromer complex has gained recognition in the field of recycling of major solute carriers, cell surface receptors, and other transmembrane proteins. However, the mechanisms driving SNX27 mediated endosomal recycling of proteins remains largely unknown. 

Interactions between the intracellular compartments and the extracellular environment need to be closely monitored in order to maintain cellular homeostasis [12]. Therefore, improper sorting of endocytosed cargo can disrupt the balance between the cell surface and intracellular expression levels of transmembrane proteins, which may contribute to disease progression and pathophysiology, if left unchecked.

Emerging evidence suggests a potential role of SNXs in human cancers, although the research is still limited. This review is aimed towards summarizing scientific evidence supporting and providing an update on the current molecular understanding of SNX27, its associations with the sorting and transport of internalized endosomal cargo, and its implication in cancers.

## 2. Overview of the Sorting Nexin Family of Proteins

Since the discovery of the first sorting nexin, a total of 10 family members have been identified in yeast and 33 members in mammals [13]. Typically, the sequence length of SNX proteins range between 400–700aa with a common PX domain, which is approximately 100–130aa long [14]. Since PX domains have high binding affinities to phosphatidylinositol 3-monophosphates (PtdIns3P), SNX proteins are attracted to organelle membranes rich in these phospholipids, such as the endosomal membrane [15]. However, some members of the SNX family have additional structural domains that have shown to regulate their target specificity, intracellular localization, as well as membrane conformation, which also gives rise to various SNX subfamilies (see Table 1) [2]. 

One of the most critical physiological roles of SNX proteins that has been studied so far is their ability to mediate the trafficking of endocytosed cargoes from the compartments of early endosomes to either the TGN or the plasma membrane [16]. However, it has been established that the successful recycling and sorting of endocytosed cargoes is dependent on the association of SNX proteins with the retromer complex, a heterotrimer of VPS35, VPS26 and VPS29 subunits [17]. In the mammalian system, three different SNX-retromer associations have been identified which include the SNX3-, SNX-BAR-, and SNX27-retromer complexes [18]. 

SNX3 belongs to the SNX-PX subfamily of SNX proteins as it contains only the PX domain. The association of SNX3 with the retromer subunit, VPS26, has been shown to be crucial for cargo recognition, binding, and retrograde recycling to the TGN [10]. This is due to the fact that the SNX3/VPS26 interaction forms a binding site at its interface for cargoes enriched in L-M-V domain, while the SNX3-PX domain exclusively interacts with the cytosolic face of early endosomes [19]. Some of the SNX3-retromer cargoes that have been identified include Wntless, a Wnt sorting receptor [20], and Dmt-II metal ion transporter [21].

In mammals, up to twelve members of the SNX-BAR subfamily have been identified which possess a Bin/Amphiphysin/Rvs (BAR) domain in addition to the highly conserved PX domain [22]. Among these twelve members, there are high structural similarities between SNX1 and 2, as well as SNX5 and 6, which form a functional SNX1/2:SNX5/6 heterodimer, representing their yeast counterpart Vps5:Vps17 [13]. Further analysis revealed that SNX1, the first SNX protein to be identified in mammals, binds to the retromer and mediates the formation of the SNX-BAR-retromer complex consisting of the SNX1/2: SNX5/6 heterodimer subunit [23]. This association further promotes the membrane remodeling and tubule formation necessary for promoting the retrograde transport of selected cargoes, such as the mannose-6-phosphate receptor, to the TGN [24].

SNX27 was identified as another retromer associated SNX protein belonging to the SNX-FERM subfamily. Members of this subfamily contain a PX domain and a C-terminal 4.1/ezrin/radixin/moesin (FERM) domain [25]. The mechanisms of SNX27-mediated endocytic recycling of proteins has not been fully elucidated yet; however, the current understanding suggests that the SNX27-retromer complex assists in the direct trafficking of cargo from the endosomal compartments to the plasma membrane, and that it requires an association with the SNX-BAR subunit [26]. Furthermore, SNX27 is a unique member of the SNX family as it is the only known SNX protein carrying an additional N-terminal post-synaptic density 95/discs large/zonula occludens-1 (PDZ) domain, and current models suggest that the SNX27-retromer complex recognizes and binds to target proteins in a PDZ-domain dependent manner [3]. 

In addition to their primary role in endocytic recycling, some SNX proteins are involved in the process of endocytosis and lysosomal degradation. SNX9 and SNX18 carry an N-terminal SH3 domain in addition to the C-terminal BAR domain [27]. While SNX9 can directly bind to dynamin-1/2 and regulate clathrin-mediated endocytosis, SNX18 has been particularly shown to compensate for the loss of SNX9 during endocytosis of transferrin receptors [28]. Meanwhile, SNX1, SNX6 and SNX11 have shown to assist in the sorting and lysosomal degradation of epidermal growth factor receptor (EGFR) [23], tumor suppressor p27Kip1 [29], and TRPV3 [30], respectively. 

## 3. Structure and Function of SNX27

### 3.1. SNX27 and Its Special Domains

As shown in Figure 1, SNX27 contains three distinct domains: a central PX domain, a C-terminal FERM domain, and an N-terminal PDZ domain (only in SNX27) [2] (Figure 1). Despite carrying a common PX and FERM domain, SNX27 behaves functionally distinct from the other members of the SNX-FERM subfamily [25]. While both SNX17 and SNX31 depend on their FERM domain and have strong affinity towards targeting and recycling cargoes carrying a NPxY/NxxY motif, SNX27 mediates endocytic recycling of cargoes in a PDZ-domain dependent manner [25]. The SNX27-FERM domain associates with an N-terminal DLF motif found on the SNX1/2 homodimer, which has not been reported in either SNX17 or SNX31 [25]. This was further supported by structural analysis data, which reported very low sequence identity between SNX27-FERM domain compared with that of SNX17 and SNX31 [25]. Therefore, the FERM domain has a distinct role in SNX27 than in the other SNX-FERM members, such that it only promotes SNX27 interaction with the SNX-BAR proteins. This structure mediates the recruitment of SNX27 to the membranes of early endosomes and initiates membrane remodeling to assist in the endocytic recycling and trafficking of PDZ-domain containing cargoes [31].

SNX27 was initially discovered in the brain tissue where it was found to be interacting with 5-HT4.a/R, a G-protein coupled receptor (GPCR) [32]. Structural analyses later revealed that 5-HT4.a/R carries a C-terminal class-I PDZ-binding motif and its association with SNX27 was found to be dependent on its interaction with the PDZ-domain [33]. Therefore, the canonical basis for the PDZ domain to recognize and bind to cargoes was predisposed to the presence of a class-I PDZ-binding motif [S/T]-x-Φ (where x represents any amino acid and Φ represents any hydrophobic amino acid) at the C-termini, often present in the cytosolic tails of transmembrane proteins [34]. However, presence of certain sequences upstream of the C-terminus PDZ-binding motif was also shown to enhance binding specificity [35]. Studies have revealed that acidic side chains are often found alongside the PDZ-binding motifs, however several SNX27 cargoes have replaced these upstream acidic side chain sequences with conserved serine/threonine phosphorylation sites which was later shown to mimic the acidic side chain residues, thus promoting binding affinity in a similar manner [35]. With the advent of proteomics and sequence quantitative analysis, over hundreds of transmembrane proteins have been identified to be interacting with SNX27-PDZ domain. These include β-2-adrenergic receptor, GLUT1 glucose transporter, and the Menkes copper transporter ATP7A [36]. 

### 3.2. SNX27 Mediated Endosomal Recycling of Proteins

Endocytosed transmembrane proteins require to go through a complex network of the endomembrane system that is subdivided into the early, late, and recycling endosome vesicles. While the cargo sorted in late endosomes proceed for lysosomal degradation, those present in the recycling endosomes further interact with the components of the secretory pathway in order to be transported to either the TGN or delivered back to the cell surface [6]. Therefore, improper sorting of endocytosed cargo can disrupt physiological homeostasis and give rise to various health problems and diseases [7]. 

Sorting and recycling of the transmembrane proteins from early/recycling endosomal compartments directly to the plasma membrane is heavily dependent on the association of the SNX27/SNX-BAR/retromer complex on Rab4-positive endosomes [37]. In this model, the PX domain is critical for the recruitment and binding to the endosomal membrane; the FERM domain recruits SNX-BAR heterodimers which initiates membrane remodeling and tubule formation necessary for transporting the cargo [1], and the PDZ domain binds to VPS26 retromer subunit and acts as the cargo adaptor [36]. Studies have shown that the disruption of either of these components impairs SNX27 dependent cargo binding and trafficking. 

While the PX domain drives SNX27 membrane recruitment, structure-based analyses have shown that the synergistic interaction of the FERM domain with the endosomal membranes also enhances localization of the SNX27-retromer complex to the endosomes, as disrupting the FERM domain decreased the affinity between SNX27 and the endosomal compartments [25]. 

The actin remodeling WASH complex, which consists of the WASH1, WASHC3, WASHC4, WASHC5, and FAM21 subunits [38], was identified as another SNX27-retromer interacting partner, which synergistically binds to the SNZ27-FERM domain as well as the VPS35 subunit of retromer via FAM21 [39]. Upon its recruitment, the WASH complex has shown to initiate the formation of F-actin filaments on the endosomal membranes which promotes trafficking of cargo from endosomes to the plasma membrane directly [40]. This has been particularly observed in the SNX27 PDZ-domain-dependent cell surface recycling of GLUT1, as disrupting the binding of the WASH component to the SNX27-retromer complex directed the recycling of GLUT1 to the TGN instead of the plasma membrane [26]. Therefore, this interaction may be critical for the assembly and/or regulation of the SNX27-retromer complex.

Among the many identified PDZ-domain-containing cargoes dependent on the SNX27-retromer-mediated endosome-to-plasma membrane recycling (see Table 2), β-2-andrenergic receptor was the very first transmembrane protein [36]. Some other cargoes include Ras, a monomeric small GTPase [25]; zonula occludens 2 (ZO2), an epithelial tight junction protein [36]; and AMPA receptor [41]. More recently identified cargoes include OTULIN, a deubiquitinating enzyme, which contains a class-1 PDZ-binding motif [42]. However, via X-ray crystallography it was observed that OTULIN has a high affinity to the PDZ-VPS26 binding site in addition to the canonical PDZ domain-binding motif interaction site [42]. SNX27 interactome analysis also revealed NHE3, which depends on the PDZ-domain interaction with SNX27 for its recycling from early endosomes to the plasma membrane [43]. In addition to maintaining NHE3 surface expression levels, SNX27 was also found to be necessary for brush border stability as SNX27 deletion in intestinal epithelial cells reduced NHE3 basal activity. Similarly, DRA was found to be dependent on SNX27 for its recycling to the apical plasma membrane, as shown in CaCo2 intestinal epithelial cells [44]. These studies have revealed a novel function of SNX27 mediated endocytic recycling of transmembrane proteins in the gastrointestinal tract which may be of importance in studying disease pathogenesis.

In contrast to the canonical mechanism of class-1 PDZ-binding motifs necessary for interaction with SNX27, it was recently observed that MT1-MMP, which lacks a class-1 PDZ-binding motif on its C-terminus, is still able to bind to the SNX27-retromer complex and is recycled to the cell surface in a SNX27-retomer-dependent manner [50]. Structural analysis revealed that this interaction is mediated via the DKV motif present in the cytosolic tail of MT1-MMP, which has shown to possess the features similar to a class-III PDZ-binding motif, however further biophysical studies are deemed necessary to fully understand this association [62]. This observation provides novel insights into the possibilities of non-canonical mechanisms of SNX27 PDZ-domain interactions with its cargoes.

Overall, SNX27’s interacts with various transmembrane proteins for the maintenance of their cell surface levels and activity. It suggests that loss of SNX27 may lead to an array of cellular dysfunction, which has been reciprocated in vivo as SNX27 whole-body knockout mice have been shown to be embryonically lethal [63].

### 3.3. SNX27 and Activation of T-cells

SNX27 is found to be expressed in almost every cell type, based on the data from Human Atlas (https://www.proteinatlas.org/ENSG00000143376-SNX27/tissue, accessed on 21 December 2022). Primarily, SNX27 was studied in the brain–nervous system and related diseases. However, we believe its role in other organs could be equally critical.

SNX27 was found to be localized at the immune synapse (IS) in a T-cell receptor (TCR) activation dependent manner [64]. Cells of the immune system give a great example of an intricate molecular network that depends on the endocytic recycling pathway to maintain constant communication between the intercellular components and the extracellular environment. The plasma membrane of T-cell lymphocytes expresses several important receptors, including TCR, which is necessary for the recognition of antigens presented by an antigen-presenting cell (APC) [65]. The binding of the TCR and a recognized antigenic peptide triggers a rapid morphological change within the T-cells causing actin remodeling and vesicular polarized trafficking of organelles towards the T cell-APC interface, thus forming the immune synapse (IS) [66]. This event is critical for sustaining intact communication between T-cell and the APC which ensures the activation of T-cell and downstream immune responses [67]. Extensive studies revealed that T-cells maintain surface level TCR expression and other signaling components necessary for IS via endosomal recycling pathways [68,69]. 

Under unstimulated conditions, SNX27 is predominantly expressed on the PtdIns[1]P-rich membranes of early and recycling endosomes, which then rapidly polarizes towards the IS upon TCR activation [34]. Structural studies have revealed that this redistribution is mediated by the interactions between the PX and FERM domains of SNX27 with PtdIns[1]P and PtdIns(4,5)P2/PtdIns(3,4,5)P3-enriched membranes, respectively [70,71]. This was further supported by Tello-Lafoz et al., as they showed that disruption of the FERM domain impaired SNX27 spatial distribution during IS initiation [64]. 

Proteomic analysis of the SNX27 interactome in IS forming activated T-cell lymphocytes has not only confirmed the participation of retromer and WASH complexes in SNX27-mediated polarized trafficking, but also revealed several SNX27 interacting cargoes that are presented at the IS in a PDZ-dependent manner, e.g., diacylglycerol kinase-ζ (DGKζ) [34,72]. TCR activation leads to the generation and accumulation of diacylglycerols (DAGs) at the IS which facilitates the recruitment of other signaling proteins involved in IS formation and maintenance [73]. While DGKζ is a negative regulator of DAG, its association with SNX27 prevents degeneration of DAG, thereby regulating IS stability via SNX27-mediated DAG metabolism [34]. Likewise, disrupting the SNX27-DGKζ interaction in activated T-cells affected downstream signaling pathways, as indicated by increased ERK phosphorylation and NF-κB hyperactivation upon either SNX27 or DGKζ silencing [34,74,75]. However, silencing of SNX27 does not affect protein expression levels of DGKζ, which suggests that the association of DGKζ with SNX27 is only necessary for its spatial distribution and trafficking during IS formation [75]. 

The identification of DGKζ and its association with SNX27 provides novel insights into DAG associated T-cell activation. It also suggests the tissue specific role of SNX27 in immune system. Further investigation is required to determine the extent of SNX27-mediated regulation of IS assembly and participation of other SNX27-interacting cargoes in this process. Underlying the importance of SNX27 mediated endocytic recycling of a wide array of transmembrane proteins with distinct functions, SNX27 is necessary for proper functioning of human health [76].

### 3.4. SNX27 in Neurodegenerative Disorders

The PDZ binding motif is most commonly found in proteins involved in excitatory synapses, and thus located within regions of postsynaptic densities in the neurons [77]. In the brain itself, SNX27 is found to be localized primarily within dendrites and has been shown to regulate synaptic plasticity [78]. SNX27 was first identified in the brain in an experiment wherein metamphetamine induced stimulation of dopamine receptors caused an upregulation of SNX27 [79]. Additionally, in vivo studies have shown that SNX27 is also critical for postnatal growth and survival as SNX27-/- mice die shortly after birth [56]. Therefore, dysregulated SNX27 functioning has been reported in several neurological and degenerative diseases, such as Alzheimer’s disease. 

Multiple studies have shown enlargement of early endosomes and multivesicular bodies in brain tissues and neurons isolated from human as well murine models of Alzheimer’s disease (AD), overall suggesting defects in endocytic trafficking and recycling pathways [80,81]. SNX27 loss-of-function proteomic analysis has also demonstrated that SNX27 mediates recycling of internalized AD-related protein APP, as deletion of SNX27 decreased cell surface expression levels of APP [60,82]. However, the authors of this study were unable to detect direct binding of SNX27 to APP, suggesting the involvement of an intermediate molecule promoting SNX27/APP interaction [3]. Previously, an intracellular sorting receptor, SorLA, was identified as an APP binding protein, and reduction in SorLA expression levels caused APP cellular redistribution [83]. Therefore, functional studies looking into the interaction between SNX27, APP and SorLA were conducted, and the results revealed that SorLA binds to APP as well as SNX27 and forms a ternary structural protein complex, thereby acting as the molecular link in SNX27 mediated trafficking and recycling of APP [60]. 

SNX27 has also been reported to contribute towards impairment in neuronal and learning abilities in Down’s Syndrome (DS) [77]. The overall expression levels of SNX27 are shown to be downregulated in brain tissues from human patients and murine models of DS. Interestingly, rescue studies in DS mouse model have shown that overexpression of SNX27 reverses or corrects DS-related cognitive and synaptic impaired phenotypes [77]. Further analysis revealed that expression of SNX27 is regulated by C/EBPβ, a transcription factor targeted by miR-155. It was identified that miR-155 is a micro-RNA encoded on chromosome 21; therefore, trisomy of chromosome 21 in DS drives up miR-155 which causes silencing of C/EBPβ, thereby downregulating SNX27 [77]. More recently SNX27 was shown to recycle the myelination-related protein, GRP17, which is crucial for the differentiation and maturation of oligodendrocytes [61]. Improper functioning and distribution of oligodendrocytes has been reported in human and mouse DS as it leads to abnormalities in the white matter of the brain which increases defects in cognitive and motor skills [84]. Therefore, these studies suggest an underlying mechanism of SNX27 mediated neuropathogenesis of DS through oligodendrocyte dysfunction.

Lastly, SNX27 has been shown to recycle AMPA and NMDA glutamate receptors [56,78]. Glutamate serves as the primary neurotransmitter in the brain and therefore plays a critical role in mediating cognitive function and excitatory synapses, both of which promote learning and motor abilities [85]. Consequently, heterozygous SNX27-/+ mice appear to have learning and memory disabilities associated with a reduction in cell surface expression of AMPA or NMDA receptors [63]. Therefore, dysregulated expression of SNX27 can lead to seizures and epilepsy among a plethora of other neurodegenerative diseases [77].

## 4. Identifying the Novel Role of SNX27 in Cancer

Increasing evidence suggests that SNX27 may mediate cancer development and/or progression as it influences distinct protein–protein interactions, membrane remodeling, and cell surface expression of several important tight junction proteins and receptors implicated in cell signaling pathways (see Table 3) [35]. Human genome database analysis reveals that SNX27 is highly expressed in several cancers, in particular invasive breast cancer, which is inversely correlated with overall patient survival [86].

We have previously demonstrated that loss of SNX27 dramatically reduces tumor growth and proliferation in breast cancer cells, which was further confirmed via mouse xenograft models [87]. We further reported a novel role of SNX27 in reducing aggressiveness and invasive capacity of the cancer cells via the modulation of EMT marker Vimentin, and cell–cell junction markers, such as Claudin-5 and E-cadherin. 

Another study investigating the associations of SNX27 and breast cancer metastasis revealed that SNX27 directly interacts with MT1-MMP, a key enzyme recruited by invadopodia for degradation and remodeling of the extracellular matrix [50]. The in vitro analysis demonstrated that the association of the SNX27-retromer complex is critical for the recycling of MT1-MMP to the cell surface, as deletion of either component resulted in the accumulation of MT1-MMP in late endosomes. It is also of interest to note that, unlike the other known SNX27 binding partners, MT1-MMP lacks the PDZ-binding motif. Instead, the cytosolic tail of MT1-MMP harbors a DKV motif that closely resembles a Class-III PDZ-binding motif sequence of X[DE]Xϕ, where X represents any amino acid and ϕ represents a hydrophobic residue [88]. Therefore, this study revealed a new binding partner of SNX27 harboring a Class-III PDZ-binding motif. 

Several studies have revealed protein candidates from the GPCR superfamily which interact with SNX27. These include the β2-andrenergic receptor [1], parathyroid hormone receptor (PTHR) [47], metabotropic glutamate receptor 5 (mGluR5) [48], G protein-coupled inwardly rectifying potassium channel subunits (GIRK 1 and 2) [45,46], and frizzled receptor 7 (FZD7) [49]. These GPCRs are known to be involved in tumorigenesis. For example, mGluRs have been heavily reported in neuroblastoma and gliomas [89], with mGluR5 particularly driving tumorigenesis as its inhibition has shown to facilitate hypoxia-induced cell death in gliomas [90], and GIRKs are highly expressed in over 70% of non-small cell lung cancer patients which correlates with more aggressive tumors [91]. Similarly, β2-andrenergic receptor [92] and PTHR [93] have been reported to drive tumorigenesis via EMT regulation, whereas FZDs mediates tissue homeostasis and the canonical Wnt signaling pathway known to be regulated in several cancers [94]. However, there are limited studies on the tissue-special roles of SNX27 in various cancers. It is important to identify cancer-associated proteins that may interact with the PDZ-domain in a SNX27-dependent recycling and such proteins could be targeted to impair tumorigenesis.

SNX27 is involved in cellular nutrient uptake as it directly interacts with and recycles GLUT1, facilitator of glucose transport across the plasma membrane, which promotes cell growth and survival [5]. The mTORC1 activation has also been indicated to be regulated by SNX27 [3]. Although the direct mechanisms remain to be poorly understood, these studies suggest that SNX27 associations may influence cancer cell proliferation and survival by regulating intracellular energy levels. 

**Table 3 cancers-15-00070-t003:** SNX27 associated proteins reported in cancers.

Target	Type of Cancer	Ref.
β2-ar	Breast, gastric, pancreatic, squamous cell carcinoma	[95]
GIRK2	Breast, lung, hepatocellular carcinoma	[91]
PTHR	Breast, colorectal, prostate cancer	[96]
mGluR5	Gliomas, melanoma, prostate, hepatocellular carcinoma	[97]
FZD7	Prostate, glioma, gastric, lung, hepatocellular carcinoma	[94]
ASCT2	Breast, lung, prostate cancer	[98]
MRP4	Leukemia, liver, lung cancer	[99]
SSTR5	Pituitary adenomas, prostate cancer	[100]
CASP	Breast, pancreatic, hepatocellular carcinoma	[101]
NR2C	Pancreatic cancer	[102]
AQP2	Renal cancer	[103]
β-Pix	Breast and colorectal cancer	[104]
mTOR	Several cancers: mediates cancer cell survival under stress	[105]
GLUT1	Several cancers: mediates nutrition uptake in cancer cells	[106]
DGKζ	Several cancers: mediates immune checkpoint in cells	[107]

Taken together, SNX27 may potentially exhibit its roles and regulate different stages of tumorigenesis, as we proposed in the Graphic abstract. Observations from previous studies indicate a potential therapeutic strategy of targeting SNX27 for cancer treatment, however further research is required to fully elucidate the role of SNX27 in tumorigenesis.

## 5. Roles of other SNX Members in Solid Cancers

Owing to the canonical physiological roles of the SNX family of proteins in mediating the endocytic pathway and intracellular protein trafficking, several SNXs have been involved in crucial intracellular signaling cascades, such as the Notch, EGFR, and Wnt signaling pathways, to name a few [108]. These pathways are known to mediate varying degrees of tumorigenicity in different types of cancer cells. Although limited, the investigation on the roles of other SNXs in different cancers has been summarized below: 

A study by Zhan et al. reported that the overall mRNA and protein expression levels of SNX1 was lower in gastric cancer patients, which also correlated with poor patient survival [109]. As in vitro overexpression and knockdown studies were conducted, they observed that exogenous expression of SNX1 reduced the proliferative and migration capacity of the cancer cells. As a result, they observed an upregulation of the tumor suppressor marker E-cadherin, and downregulation of the EMT marker Vimentin, whereas the effects were reversed upon reducing the levels of SNX1. These observations are in line with another study which suggested the dependence on SNX1 for the recycling of internalized E-cadherin [110]. Therefore, a novel role for SNX1 as a putative tumor suppressor marker in gastric cancer, possibly via EMT regulation, has been suggested. 

The first report of SNX5 in human cancers was identified in thyroid cancer where the expression levels of SNX5 decreased through the progression of well-differentiated to poorly differentiated tumors [111], the latter being more aggressive [112]. However, recent investigation revealed differential expression levels of SNX5 in head and neck squamous cell carcinoma where higher expression levels of SNX5 were correlated with tumor progression, aggressiveness, and poor patient survival [113]. Using mouse xenograft models, investigators revealed that loss of SNX5 reduced tumor burden by decreasing the expression levels of the anti-apoptotic protein Mcl-1 [114]. It suggests a SNX5 regulation of apoptosis in the progression of head and neck squamous cell carcinoma. 

SNX6 has been reported in pancreatic cancer and implicated in regulating EMT and invasiveness of cancer cells [115]. Loss of SNX6 resulted in the downregulation of N-cadherin and ZEB1, a mesenchymal gene and transcriptional factor, respectively, whereas upregulating E-cadherin resulted in the inhibition of the migration and invasive capacity of the cancer cells. Complementing these observations, SNX6 was highly expressed in pancreatic cancer patients and correlated with poor prognosis. 

A series of studies published by Bendris et al. revealed that SNX9 is differentially expressed through different stages of breast cancer, which can affect the functions of SNX9 as executed in different cancer cells. However, the exact mechanism regulating this phenomenon remains unclear [116]. It was also reported that SNX9 contributes to the regulation of invadopodia formation and maturation in an inverse manner, as it interacts with RhoA and TKS5, respectively. Additionally, SNX9 mediated the endocytosis of MT1-MMP metalloproteinase, a crucial component for extracellular matrix degradation [117], thereby suggesting a possible role of SNX9 in regulating cancer cell invasiveness. 

SNX10 deficiency promotes the initiation and progression of colorectal cancer in mice. Consistent to these observations, SNX10 expression levels were found to be much lower in human colorectal cancer tissues. Further analysis into the mechanism of SNX10 driven colorectal tumorigenesis revealed that loss of SNX10 activates chaperone-mediated autophagy (CMA) and increases the degradation of tumor suppressor p21 in a CMA dependent manner [118]. This was confirmed via in vitro analysis as well as mouse models, which showed an increase in tumor cell proliferation and viability upon SNX10 downregulation, which can be reversed upon exogenously increasing the expression levels of SNX10. Another mechanism underlining SNX10 mediated colorectal cancer progression was suggested through autophagy. SNX10 deficiency disrupts the autophagic degradation of proto-oncogene SRC, which in turn activates the STAT3 pathway, thus promoting colorectal cancer initiation and progression [119]. Interestingly, aberrant expression of SNX10 has been reported in hepatocellular carcinoma. Cervantes-Anaya et al. reported that the mRNA and protein levels of SNX10 are inversely expressed in the tumors of hepatocellular carcinoma which led on to the identification of microRNA-30d which was found to negatively regulate the translation of SNX10 protein levels [120]. Therefore, not only has SNX10 been deemed as having tumor suppressor roles in colorectal cancer, it has also been suggested to be a putative marker for hepatocellular cancer with tumor suppressive potential likely regulated by a microRNA.

## 6. Discovering SNX Proteins in Liquid Cancers

As we have described above the incidence and roles of SNX proteins in various solid tumors, in this section we will discuss the reports of SNX family members particularly in liquid cancers, such as lymphoma, leukemia, and myeloma (Table 4 provides a summary of known reports identifying SNXs in various cancers). SNX2 was the first member of the SNX family to be reported in acute myeloid leukemia (AML) by the group of Fuchs et al. With the help of screening a human kidney library, this group identified SNX2 as an interacting partner of the formin-binding protein 17 (FBP17), a known MLL-fusion protein found in AML patients [121]. They also observed co-immunoprecipitation of SNX2 with EGFR, which is known to be significantly upregulated in AML [122]. Therefore, this study suggested that SNX2 can alter AML development by regulating the EGFR pathway in a FBP17-MLL dependent manner. Later in 2011, Ernst et al. identified SNX2 as a novel gene fusion partner to ABL1 in B-cell acute lymphoblastic leukemia (ALL) [123]. Follow up research then revealed that the SNX2-ABL1 fusion gene is directly and functionally involved in the pathogenesis of ALL [124]. Additionally, the gene expression profile of ALL cells harboring a SNX2-ABL1 fusion transcript was similar to that of BCR-ABL1, a previously identified prognostic markers for ALL [125]. Furthermore, in vitro analysis using murine Ba/F3 cells showed that SNX2-ABL1 protein expression, similar to BCR-ABL1, transformed the cells such that they were able to proliferate in an IL-3 independent manner [126]. However, when comparing responses to kinase inhibitors, imatinib and dasatinib, it was observed that murine Ba/F3 cells carrying the SNX2-ABL1 fusion were less sensitive to the treatment as the rate of apoptosis was much lower in the SNX2-ABL1 versus BCR-ABL1 expressing cells. Therefore, further research is deemed necessary to identify selective tyrosine kinase inhibitors in response to SNX2-ABL1 specific chimeric fusion protein expressed in ALL patients. 

Similarly, a study has implicated a therapeutic potential of SNX10 in treating B-cell non-Hodgkin lymphoma (B-NHL) [127]. This study highlighted the potent anti-tumor activity and sensitivity of apilimod against B-NHL, as displayed in both in vitro as well as in vivo models of the disease. In order to determine the underlying genomic factors that attributes B-NHL cells sensitive towards apilimod, they performed a genome wide screen using CRISPR and identified SNX10, among other candidates, whose loss of function resulted in B-NHL resistance to apilimod. They also observed that lack of SNX10 in B-NHL cells impairs maturation of lysosomes, disrupts autophagy clearance, and promotes activation of transcription factor (TFEB). Overall, this study identified a novel role of SNX10 in B-NHL drug sensitivity towards apilimod [127].

Although direct validation studies have not been conducted yet, there are speculations that SNX9 may also play a role in the development of liquid cancers. A recent siRNA mediated knockdown study has reported that the loss of SNX9 dramatically reduces internalization of ADAM9 [128]. Previous studies have shown that ADAM9 is a pro-tumorigenic protein and promotes aggressive phenotype in several cancers, including myeloma [129]. Additionally, depletion of SNX9 resulted in the increase of ADAM9 cell surface expression levels and its functional activity as indicated by an increase in the shedding of its substrate EphB4. Therefore, this study provides evidence towards the tumor-suppressor roles of SNX9 and also indicates towards its potential in the prevention of liquid cancers, among others, in an ADAM9-dependent manner.

SNX27, the most unique member of the SNX family, has also been identified to play a role in leukemia. An RNAi screen conducted in primary AML cells reported that SNX27 promotes AML tumorigenesis and may serve as a potential prognostic marker, since a lack of SNX27 dramatically reduces tumor cell growth and viability [130]. More recently, it was demonstrated that Tax-1 interacts with SNX27 in a PDZ-dependent manner [131]. Tax1 is a regulatory gene required for the replication and pathogenesis of adult T-cell leukemia causing virus, HTLV-1 [132]. This study showed that Tax-1 overexpression reduces GLUT1 cell surface levels and increases its lysosomal degradation, an effect previously demonstrated by SNX27 loss of function studies [3]. Additionally, GLUT1 is a well-known HTLV-1 receptor molecule previously implicated in viral fusion, shedding, and infectivity [133]. Therefore, this study describes a novel role of SNX27 in the pathogenesis of leukemia causing virus as maintenance of GLUT1 cell surface levels via Tax-1/SNX27 interaction provides novel insight into post-infection regulation of HTLV-1 receptor molecules.

**Table 4 cancers-15-00070-t004:** Summary of SNXs reported in various cancers.

Member	Type of Cancer	Associated Protein	Mechanism of Action	Ref.
SNX27	Acute myeloid leukemia	Unknown	SNX27 deletion reduces tumor cell viability and proliferation	[130]
	Adult T-cell leukemia	Tax1	Tax-1 interacts with SNX27 to regulate GLUT-1 and pathogenesis of ATL causing virus HTLV-1	[131]
	Breast cancer	MT1-MMP	SNX27 directly recycles MT1-MMP and enhances tumor invasiveness	[50]
	Breast cancer	E-cadherin, Vimentin	SNX27 deletion regulates EMT markers and reduces tumor cell viability and proliferation	[87]
SNX1	Gastric cancer	E-cadherin	Upregulation of SNX1 reduces tumor cell invasion and aggressiveness	[109,110]
SNX5	Thyroid cancer	Unknown	SNX5 expression decreases with tumor progression	[111,112]
	Head and neck squamous cell carcinoma	Mcl-1	SNX5 deletion induces apoptosis	[113]
SNX6	Pancreatic cancer	ZEB-1, E-cadherin, N-cadherin	SNX6 deletion regulates EMT markers and reduces tumor cell aggressiveness	[115]
SNX9	Breast cancer	MT1-MMP	Upregulation of SNX9 promotes MT1-MMP endocytosis and reduces tumor invasiveness	[117]
	[ADAM9 implicated in leukemia and lymphoma]	ADAM9	SNX9 negatively regulates ADAM9 surface levels	[128]
SNX2	Acute myeloid leukemia	FBP17, EGFR	SNX2 interacts with prognostic FBP17/MLL fusion protein and regulates EGFR signaling	[121]
	B-cell acute lymphoblastic leukemia	ABL1	Expression of SNX2/ABL1 fusion protein reduces drug sensitivity	[123]
SNX10	B-cell non-Hodgkin lymphoma	Unknown	Deletion of SNX10 reduces drug sensitivity	[127]
	Colorectal Cancer	p21, SRC	Overall downregulated; loss of SNX10 decreases p21 tumor suppressor and increases SRC	[118,119]
	Hepatocellular carcinoma	miR-30d	SNX10 is silenced by miR-30d oncomir which promotes cancer progression	[120]

## 7. Future Perspectives and Conclusion

A lot of research has been focused on unravelling the fundamental roles of SNXs in synchronizing the various stages of endosomal cargo selection and transport [2]. The SNX family comprises members that are functionally and structurally distinct from one another, which influences their involvement in endosomal cargo sorting and trafficking mechanisms [13]. Genome wide analysis provides a helpful platform to identify differential expression levels of target SNXs in various cancers; however, identifying the underlying mechanisms driving the correlation between SNXs and cancer remain to be properly investigated. Determining the importance of individual SNX protein in maintaining intracellular protein homeostasis will help to understand the etiology of cancers. 

SNX27 is the most unique member of the SNX family of proteins which mediates endocytic recycling of several critical transmembrane proteins in a PDZ-domain dependent manner [1]. However, research is still ongoing to identify and validate SNX27 binding cargoes that may have critical roles in maintaining human health and regulating diseases. Additionally, the non-canonical mechanisms and functions of SNX27 remain unexplored. Therefore, biological and structural analysis into understanding the mechanisms of SNX27-retromer-mediated sorting and transport of endocytosed proteins may provide novel insights into the otherwise unknown role(s) of SNX27. 

Several studies have implicated SNX27 in regulating various cancers, particularly breast cancer [87]. This has been supported by genome wide database analysis, in vitro gene regulation studies, as well as in vivo xenograft models [87]. We reported the reduced SNX27 in suppressing autophagy in colonic cancer cell lines [134]. However, in order to fully understand the molecular role of SNX27 in pathways involved in the individual steps of tumorigenesis, detailed research is required that focus on using appropriate and innovative techniques and experimental cancer models to better mimic the tumor microenvironment in vivo. Microarray analysis has helped in the identification of several PDZ-binding motif carrying proteins that are yet to be experimentally validated [88]. Various GPCRs have been implicated in tumorigenesis as well as associations with SNX27, however the exact mechanism driving SNX27 mediated interaction in cancer has not been explored [86]. Therefore, future studies are required to study and demonstrate the crucial role of SNX27 in tumorigenesis. New insights into the roles of SNX27 in various cancers will devise and strategize novel SNX27 associated therapeutic targets for prevention and treatment of these challenging diseases.

## Figures and Tables

**Figure 1 cancers-15-00070-f001:**
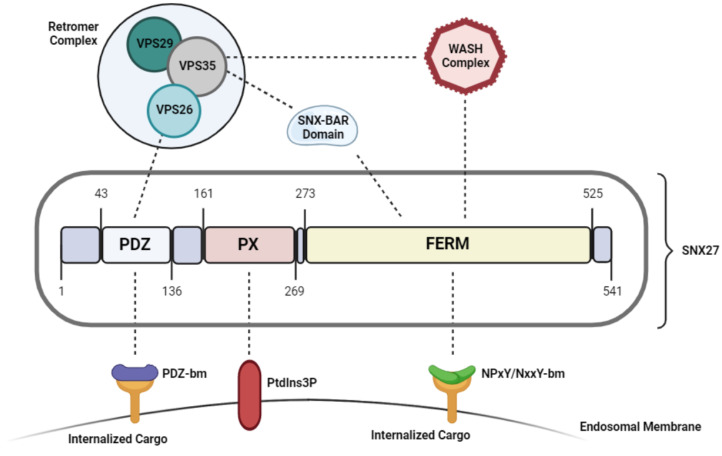
Structure and functional domains of SNX27.

**Table 1 cancers-15-00070-t001:** Characteristic features of different SNX domains.

Subfamily	Members	Primary Roles	Links to Processes Indicated in Health and Disease Regulation
SNX-PX	SNX3	Assists in the recruitment of the retromer complex to early endosomes	SNX3-retromer complex distinctly carries out cargo recognition,
SNX10	binding, and retrograde recycling to the TGN and
SNX11	SNX3 is also important for neurite development
SNX12	
SNX16	SNX10 regulates autophagy
SNX20	
SNX21	SNX11 mediates lysosomal degradation of EGFR and TRPV3
SNX22	
SNX24	SNX16 regulates endosomal membrane dynamics and mediates
SNX29	trafficking between early and late endosomes
SNX-BAR	SNX1	Additional C-terminal BAR domain comprising three α- helices	SNX9 and SNX18 carry an additional SH3 domain and
SNX2	regulate clathrin-mediated endocytosis of cargoes
SNX4	
SNX5	SNX1 mediates trafficking of mGluR1 in hippocampus and
SNX6	SNX1 SNPs are found in brains of Alzheimer’s patients
SNX7	Heterodimer of SNX1:SNX2 and SNX5:SNX6 is involved in retromer mediated trafficking	
SNX8	SNX2 is overexpressed in the hypothalamus of aging mice
SNX9	
SNX18	SNX4, 6, 7, 8 regulate Aβ generation by mediating APP and
SNX30	BACE1 trafficking and lysosomal degradation
SNX32	Involved in membrane tubulation	
SNX33	SNX5 is associated with trafficking of Dopamine receptor 1
SNX-FERM	SNX17 SNX27 SNX31	C-terminal FERM domain assists in endocytic traffickingand lysosomal degradation	SNX17 and 31 target cargoes carrying a NPxY/NxxY motif and
cause lysosomal degradation in a FERM-dependent manner

SNX27-FERM domain binds with SNX1/2 homodimer to form
SNX27 carries an additional N-terminal PDZ domain that helps to sort and recycle proteins to the cell surface	the SNX27/SNX-BAR/Retromer complex

SNX27 modulates synaptic plasticity in the brain and is
associated with Epilepsy, Alzheimer’s, and Down’s Syndrome

**Table 2 cancers-15-00070-t002:** Identified binding partners of SNX27.

Target ID	Target Name	Ref.
β2-ar	β2 adrenergic receptor	[1]
5-HT4.a/R	5-hydroxytryptamine type 4 receptor	[32]
GIRK2	G protein-gated inwardly rectifying potassium 2	[45]
GIRK3	G protein-gated inwardly rectifying potassium 3	[46]
PTHR	Parathyroid hormone 1 receptor	[47]
mGluR5	Metabotropic glutamate receptor 5	[48]
FZD7	Frizzled receptor 7	[49]
GLUT1	Glucose transporter 1	[3]
ATP7A	ATPase copper transporting alpha	[3]
ASCT2	Alanine-, serine-, cysteine-preferring transporter 2	[5]
Ras	Ras GTPase	[25]
OTULIN	OTU Deubiquitinase With Linear Linkage Specificity	[42]
NHE3	Sodium (Na+)/hydrogen (H+) exchanger 3	[43]
DRA	Downregulated in adenoma	[44]
MT1-MMP	Membrane type 1 matrix metalloproteinase	[50]
DGKζ	Diacylglycerol kinase zeta	[34]
ZO-2	Zonula occludens 2	[51]
AMPA	α-amino-3-hydroxy-5-methyl-4-isoxazolepropionic acid receptor	[52]
β1-ar	β1 adrenergic receptor	[53]
SSTR5	mouse Somatostatin receptor subtype 5	[54]
CASP	Cytohesin associated scaffolding protein	[55]
NR2C	N-methyl-D-aspartate (NMDA) receptor 2C	[56]
AQP2	Aquaporin 2	[57]
β-Pix	β-PAK-interacting exchange factor	[58]
Git	G-protein receptor kinase interacting target	[58]
MRP4	Multidrug resistance-associated protein 4	[59]
SorLA	Sorting-related receptor with A-type repeats	[60]
GRP17	Gadd related protein, 17 kDa	[61]

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
