# Peer review of "Endosomal Sorting Protein SNX27 and Its Emerging Roles in Human Cancers"

_cancers, 2022, doi:10.3390/cancers15010070_

Round 1

Reviewer 1 Report

The sorting nexin (SNX) family of proteins, which have been studied in numerous tumorigeneses, mediates the trafficking and sorting of the endocytosis pathway. More than 100 cell surface proteins that are connected to several signaling pathways are thought to have a role in oncogenic signaling and have been identified as SNX binding partners. In this review, the authors discuss SNX27 function in various human malignancies. Although the paper is well written, I do have a few concerns.

Even though the authors have covered the involvement of SNX in numerous solid tumors, I suggest that the authors describe the function of SNX in liquid cancers. Also, write in a separate heading.

The table that details the characteristics of the SNX family protein structure in various malignancies should be included by the authors.

Author Response

We deeply appreciate the feedback from reviewers. The valuable inputs and suggestions allow us to further strengthen our manuscript. We have added three new Tables in the revised manuscript and addressed the respective comments as below:

Reviewer 1:

“The sorting nexin (SNX) family of proteins, which have been studied in numerous tumorigeneses, mediates the trafficking and sorting of the endocytosis pathway. More than 100 cell surface proteins that are connected to several signaling pathways are thought to have a role in oncogenic signaling and have been identified as SNX binding partners. In this review, the authors discuss SNX27 function in various human malignancies. Although the paper is well written, I do have a few concerns.”

  1. “Even though the authors have covered the involvement of SNX in numerous solid tumors, I suggest that the authors describe the function of SNX in liquid cancers. Also, write in a separate heading.”

Response: As per the recommendation, we have summarized relevant literature concerning the involvement of SNX family proteins in liquid cancers such as leukemia, lymphoma, and myeloma. We have added the information under a separate heading (#6) titled “Discovering SNX Proteins in Liquid Cancers” on page 12.

  1. “The table that details the characteristics of the SNX family protein structure in various malignancies should be included by the authors.”               

Response: We greatly appreciate this suggestion as it helps to provide a quick glance at the summary of SNX-domains, various members, and their characteristic properties relevant to the structure. We have summarized the related information under “ new Table 1. Characteristic features of different SNX domains” on page 4.

Reviewer 2 Report

The review manuscript is well written and explains in detail about the role of SNX27 protein. 

Minor comments to address:

1. The authors stated the role of SNX27 in neurodegenerative diseases. It would be best if they write an additional section about it in detail.

2. A suggestion for the section related to the role of SNX27 in cancers. I would recommend the authors to provide a table with a list of different proteins which are involved with SNX27 and what type of cancer. It will be more engaging and convenient.

3. Are there any survival curve studies on different cancer types which could indicate the survival data with high and low SNX27 expression? This data could be useful to show the impact of the protein in different cancers. 

Author Response

Title: Endosomal Sorting Protein SNX27 and Its Emerging Roles in Human Cancers

We deeply appreciate the feedback received from reviewers. The valuable inputs and suggestions allow us to further strengthen our manuscript. We have added three new Tables and addressed the respective comments as described below:

Reviewer 2 comments:

“The review manuscript is well written and explains in detail about the role of SNX27 protein.

Minor comments to address:”

  1. “The authors stated the role of SNX27 in neurodegenerative diseases. It would be best if they write an additional section about it in detail.”

Response: We thank the reviewer for finding the manuscript well written and that we were able to discuss the role of SNX27 in detail. As per the helpful suggestion, we have added a new heading (#3.4) titled “SNX27 in Neurodegenerative Disorders” on page 8 and we have summarized relevant literature concerning the involvement of SNX27 in neurological disorders.

  1. “A suggestion for the section related to the role of SNX27 in cancers. I would recommend the authors to provide a table with a list of different proteins which are involved with SNX27 and what type of cancer. It will be more engaging and convenient.”

Response: As per recommendation, we have added a new table titled “Table 3. SNX27 associated proteins reported in cancer” on page 10 entailing SNX27 binding partners that have been previously reported in human cancers.

(“Table 2. Identified binding partners of SNX27” on page 7 has been updated from the previously submitted draft of this manuscript).

  1. “Are there any survival curve studies on different cancer types which could indicate the survival data with high and low SNX27 expression? This data could be useful to show the impact of the protein in different cancers.”

Response: The reviewer’s suggestion on discussing survival analysis pertaining to SNX27 in cancers is extremely valuable to us. However, we were unable to identify any relevant literature reporting survival data on SNX27 and human cancers. In the revised manuscript, we briefly mentioned on page 8 under the heading “3.4. SNX27 in Neurodegenerative Disorders” regarding the effect of SNX27 homogeneous knockout in mice resulting in embryonic lethality.